# Production of Phytotoxic Metabolites by Botryosphaeriaceae in Naturally Infected and Artificially Inoculated Grapevines

**DOI:** 10.3390/plants10040802

**Published:** 2021-04-19

**Authors:** Pierluigi Reveglia, Regina Billones-Baaijens, Jennifer Millera Niem, Marco Masi, Alessio Cimmino, Antonio Evidente, Sandra Savocchia

**Affiliations:** 1National Wine and Grape Industry Centre, School of Agricultural and Wine Sciences, Charles Sturt University, Locked Bag 588, Wagga Wagga, NSW 2678, Australia; rbaaijens@csu.edu.au (R.B.-B.); jmniem@up.edu.ph (J.M.N.); ssavocchia@csu.edu.au (S.S.); 2Dipartimento di Scienze Chimiche, Universita’ di Napoli Federico II, Complesso Universitario Monte Sant’Angelo, Via Cintia 4, 80126 Napoli, Italy; marco.masi@unina.it (M.M.); alessio.cimmino@unina.it (A.C.); evidente@unina.it (A.E.); 3Department of Clinical and Experimental Medicine, University of Foggia, Viale Pinto 1, 71121 Foggia, Italy; 4UPLB Museum of Natural History, University of the Philippines—Los Baños, College, Laguna 4031, Philippines

**Keywords:** *Vitis vinifera*, Botryosphaeria dieback, foliar symptoms, (*R*)-mellein, qPCR, LC-MS/MS

## Abstract

Grapevine trunk diseases (GTDs) are considered a serious problem to viticulture worldwide. Several GTD fungal pathogens produce phytotoxic metabolites (PMs) that were hypothesized to migrate to the foliage where they cause distinct symptoms. The role of PMs in the expression of Botryosphaeria dieback (BD) symptoms in naturally infected and artificially inoculated wood using molecular and analytical chemistry techniques was investigated. Wood samples from field vines naturally infected with BD and one-year-old vines inoculated with *Diplodia seriata*, *Spencermartinsia viticola* and *Dothiorella vidmadera* were analysed by cultural isolations, quantitative PCR (qPCR) and targeted LC-MS/MS to detect three PMs: (*R*)-mellein, protocatechuic acid and spencertoxin. (*R*)-mellein was detected in symptomatic naturally infected wood and vines artificially inoculated with *D. seriata* but was absent in all non-symptomatic wood. The amount of (*R*)-mellein detected was correlated with the amount of pathogen DNA detected by qPCR. Protocatechuic acid and spencertoxin were absent in all inoculated wood samples. (*R*)-mellein may be produced by the pathogen during infection to break down the wood, however it was not translocated into other parts of the vine. The foliar symptoms previously reported in vineyards may be due to a combination of PMs produced and climatic and physiological factors that require further investigation.

## 1. Introduction

Grapevine is one of the most economically important crops worldwide, with approximately 71% of the world grape production being used for wine production [1]. A variety of fungal diseases threaten viticultural regions all over the world, compromising the yield and quality of the wine [2,3,4,5]. Among them, grapevine trunk diseases (GTDs), caused by one or several xylem-inhabiting fungi result in a progressive decline of vines, loss in productivity and eventually death of the vines [6].

Over the past few decades considerable knowledge has been gained on identifying GTDs, and therefore the frequency of symptoms reported due to these diseases has increased [6,7]. Recently, significant findings have been obtained on grapevine leaf stripe disease, a widely spread wood disease of Esca complex, regarding phytotoxic metabolites, symptom expression and their management [8,9]. However, the relationship of GTDs with biotic and abiotic stresses [10,11,12,13], the expression of symptoms and their effective management requires further investigation [6,14,15].

The main GTDs that threaten vineyards worldwide are Eutypa dieback (ED), Esca complex diseases, and Botryosphaeria dieback (BD), all of them caused by pathogenic fungi belonging to different families. Vines infected with ED, Esca complex, and BD usually present external symptoms such as necrotic buds, cane and trunk dieback, chlorotic leaves and leaf necrosis, and internal wood symptoms such as wedge-shaped necrotic lesions, arch-shaped necrotic lesions, brown streaking and a blackened cortex [6].

The interest in BD has increased substantially over the past decades due to increased incidence in vineyards worldwide [4,6,7,15,16]. Internal symptoms of BD include brown streaks and wedge-shaped discolorations in the wood, while external symptoms include death of the canes, shoots, and buds, stunting of shoots, delayed budburst, and foliar symptoms [4,17,18]. The appearance of foliar symptoms in grapevines infected with Esca complex and ED is frequently reported in the northern hemisphere [6,19,20,21,22]. Foliar symptoms in vines infected with BD have also been reported in Europe [17,23,24]. Nevertheless, and to our knowledge, BD foliar symptoms have not been observed in Australian vineyards to date [13].

Many foliar symptoms are usually associated with phytotoxic metabolites (PMs) produced by the causal fungi [25]. A general hypothesis suggests that PMs are translocated far from the inoculum without being detoxified, resulting in the development of foliar symptoms [26]. However, no conclusive data has been reported to support this hypothesis to date [25,27,28,29,30]. Therefore, the role of PMs in the expression of foliar symptoms requires further investigation. European isolates of Botryosphaeriaceae involved in BD have been reported to produce phytotoxins in vitro [31,32,33] and PMs produced by Botryosphaeriaceae pathogens have also been detected in infected wood with BD and Esca symptoms [34].

Considering this background and the absence of foliar symptoms in Australian vineyards, the following questions arose: (i) are the Botryosphaeriaceae spp. that infect grapevines in Australia capable of producing PMs in vitro?; (ii) are the PMs produced in vitro further produced and translocated in planta when the pathogens attack the host?; and (iii) are the PMs involved in symptom expression?

In 2016, studies were undertaken to investigate the ability of the most prevalent Australian Botryosphaeriaceae to produce PMs in vitro. These studies revealed all BD pathogens produced PMs in vitro. Some of the purified PMs were shown to cause phytotoxicity in detached leaves [35,36,37,38,39,40].

Based on these previous results, the objective of this study was to investigate the production of PMs by Botryosphaeriaceae pathogens in naturally and artificially infected grapevines, and their likelihood of translocation in woody tissue. A multifaceted approach using a combination of molecular (quantitative PCR) and analytical chemistry (LC/MS-MS) techniques were used for these investigations.

## 2. Results

### 2.1. Artificially Inoculated Vines

#### 2.1.1. Wood Symptoms

All inoculated vines did not exhibit foliar symptoms or external necrosis and showed healthy development during the entire duration of the experiment. At 6 months post-inoculation, representative vines inoculated with the pathogens, exhibited vascular staining and discoloration of the wood close to the inoculation point (Appendix A) when the bark was removed, and the trunk was cut into cross sections.

However, the lesion lengths were not significantly different between varieties or any of the inoculation treatments, including the non-inoculated negative control vines (*p* > 0.05; Figure 1). No significant differences for lesion length were observed between Botryosphaeriaceae species. All three Botryosphaeriaceae species were re-isolated from their corresponding trunk sections with necrotic lesions, while negative control vines were free of BD pathogens.

At 12 months post-inoculation, vascular staining and discoloration of wood that progressed upward and downward from the inoculation point were also observed from the longitudinal sections. The lesion lengths for all treatments were significantly longer than those vines assessed at 6 months (Figure 1). The overall lesion lengths differed significantly between varieties with Chardonnay being the most susceptible with a mean lesion length of 20.2 ± 0.8 mm, which was significantly longer (*p* = 0.01) than the mean lesion length in Cabernet Sauvignon (17.3 ± 1.0mm) (Figure 1a). The lesion lengths also varied between inoculation treatments with lesions produced by *Dothiorella vidmadera* (20.8 ± 2.7 mm), *Spencermartinsia viticola* (20.4 ± 2.8 mm), and *Diplodia seriata* (19.4 ± 0.60 mm) being significantly longer (*p* ≤ 0.05) compared to the negative control vines (13.1 ± 1.9 mm). No significant differences for lesion length were observed between Botryosphaeriaceae species. There were no significant interactions between varieties and inoculation treatments based on lesion lengths (*p* > 0.05) (Figure 1b).

#### 2.1.2. Botryosphaeriaceae DNA in Wood Tissues of Artificially Inoculated Vines

At 6 months post-inoculation, Botryosphaeriaceae DNA was detected by qPCR from wood sections with necrotic lesions near the inoculation point (IP) from vines inoculated with all three Botryosphaeriaceae species (Figure 1). The highest amount of pathogen DNA was detected from Chardonnay vines inoculated with *S. viticola*, with significantly higher amounts of Botryosphaeriaceae DNA than Chardonnay inoculated with *D. seriata* (*p* = 0.009). The amount of Botryosphaeriaceae DNA from Chardonnay inoculated with *Do. vidmadera* was not significantly different from those inoculated with *D. seriata* or *S. viticola* (Figure 2). Botryosphaeriaceae DNA detected from all Cabernet Sauvignon infected vines was significantly lower compared to those from Chardonnay, regardless of species. Significant interaction between variety and species was observed (*p* < 0.05). Botryosphaeriaceae DNA was not detected from any of the non-inoculated vines, while DNA from wood samples inoculated with *Neofusicoccum parvum* from a separate study and included as the positive control for all qPCR assays, all tested positive to the Botryosphaeriaceae DNA (data not shown).

At 6 months, Botryosphaeriaceae DNA was further detected by qPCR from non-necrotic tissues (AA’; Figure 3) for some inoculated vines. For Chardonnay, Botryosphaeriaceae DNA (69 copies) was detected from lesion-free wood sections (AA’; Figure 3) in one out of three replicate vines inoculated with *S. viticola*, while Botryosphaeriaceae DNA was not detected from any of the lesion-free wood sections (AA’; Figure 3) of vines inoculated with *D. seriata* and *Do. vidmadera*. For Cabernet Sauvignon, one out of three replicate vines inoculated with *D. seriata*, *S. viticola* and *Do. vidmadera* showed a different number of copies of Botryosphaeriaceae DNA at 40, 440 and 190 copies, respectively. For those AA’ samples (Figure 3) that were positive to qPCR, none of their subsequent BB’ sample sections (Figure 3) were positive to pathogen DNA (data not presented).

At 12 months post-inoculation, Botryosphaeriaceae DNA was further detected by qPCR from necrotic wood samples (IP; Figure 1) excised from vines inoculated with either of the three species (Figure 4). The amount of pathogen DNA across treatments increased by 10-fold compared to the amount detected at 6 months post-inoculation. The highest amount of pathogen DNA was detected from Chardonnay vines inoculated with *D. seriata* with significantly higher amounts compared to Chardonnay and Cabernet Sauvignon vines inoculated with *S. viticola* (Chardonnay *p* = 0.007; Cabernet Sauvignon *p* = 0.000007) and *Do. Vidmadera* (Chardonnay *p* = 0.003; Cabernet Sauvignon *p* = 0.00008). The lowest amount of pathogen DNA was detected from Cabernet Sauvignon vines inoculated with *D. seriata* and this was significantly lower than for all other inoculated vines. Significant interaction between variety and species was observed (*p* < 0.05) which was associated with the highest amount of Botryosphaeriaceae DNA from Chardonnay and the lowest from Cabernet Sauvignon vines inoculated with *D. seriata*. Furthermore, all asymptomatic tissues (AA’; Figure 3) regardless of treatments were negative to Botryosphaeriaceae DNA by qPCR at 12 months post inoculation.

### 2.2. Naturally Infected Vines

All wood samples with dieback symptoms, cankers and typical wedge-shaped necrosis were positive to Botryosphaeriaceae species (Figure 5, Table 1). *D. seriata* was the most prevalent species being present in three vineyards and six out of the nine vines sampled. In Hilltops, one vine was positive to *N. parvum*, while the other two vines were positive to *D. seriata* and *D. mutila*. In Tumbarumba, two vines were positive to *D. seriata* while the other vines were positive to both *D. seriata* and *N. parvum*. For the Riverina, two vines were positive to *D. seriata* while one was positive to *B. dothidea*.

### 2.3. Botryosphaeriaceae DNA in Wood Tissues of Naturally Infected Vines

All necrotic tissue samples tested positive to Botryosphaeriaceae DNA by qPCR (Figure 6). Wood samples from the Riverina vineyard contained the highest number of DNA copies that was 5-fold higher and 1.4-fold higher than the DNA from vines in Hilltops and Tumbarumba vineyards, respectively. The lowest amount of DNA was detected from all vines in Hilltops and Tumbarumba. No Botryosphaeriaceae DNA was detected in any of the non-necrotic wood samples that were collected from the same vines. Statistical analysis was not applied to the naturally infected vines because of differences in the storage and sampling of the collected wood.

### 2.4. Selection of Protocol for Extraction of PMs from Wood

Two different protocols were tested for the extraction of PMs from naturally infected wood material: (A) *n*-hexane/MeOH [34]; and (B) H_2_O/MeOH/CH_3_Cl [41]. Protocol B was less time consuming and resulted in a higher amount of organic compounds. Analysis of the organic extracts using LC-MS/MS to detect (*R*)-mellein also showed that protocol B yielded a higher amount of the target metabolite. The peak of (*R*)-mellein detected in the extract obtained with Protocol B (Appendix A, red) was 12-fold higher than the organic extract obtained with protocol A (Appendix A, green). Identification of (*R*)-mellein from extracts of symptomatic wood samples was accomplished according to its retention time, precursor ion 179.1 m/z [M + H]^+^ and fragment ions (Appendix A). Protocol B was selected for extracting PMs from inoculated vines.

### 2.5. PMs in Wood Tissues of Naturally Infected Vines

Both symptomatic and asymptomatic wood materials were analysed by LC-MS/MS after extraction, giving different results depending on the vines sampled and vineyard location. For symptomatic wood samples, only one out of three Chardonnay vines from Hilltops, two out of three Chardonnay vines from Tumbarumba and all Shiraz vines from the Riverina were positive to (*R*)-mellein. (*R*)-mellein was not detected in any of the lesion-free wood samples. A comparison of the chromatograms (Appendix A) led to the detection of a signal at a retention time of 27.44 min in the infected sample (Appendix A, red, green and purple), which was absent in the asymptomatic sample (Appendix A, black). The signal in the chromatograms was due to the (*R*)-mellein fragment ions (Appendix A). Furthermore, the area of (*R*)-mellein signals in the LC-MS/MS analysis was most significant for the Riverina vines with a high number of DNA copies of the pathogen detected in the trunk (Appendix A), indicating production of (*R*)-mellein by BD pathogens in field vines.

### 2.6. PMs in Wood Tissues of Artificially Inoculated Vines

At six months post-inoculation, no spencertoxin, protocatechuic acid and *(R)*-mellein were detected in any of the wood sections (IP, Figure 3) from non-inoculated vines. However, (*R*)-mellein was detected from the extracts of necrotic wood samples (IP) of both Chardonnay and Cabernet Sauvignon vines infected with *D. seriata* at 12 months post-inoculation (Appendix A, red) indicating that this pathogen produced (*R*)-mellein during fungal colonization. Furthermore, neither spencertoxin nor protocatechuic acid were detected from necrotic wood samples (IP) infected with *S. viticola* and *Do. vidmadera*, respectively, at 12 months post-inoculation. (*R*)-mellein was further detected from extracts obtained from the Chardonnay vine infected with *N. parvum* (DAR78998) from a separate experiment and included in the test as a positive control (Appendix A, green). The peak of (*R*)-mellein detected from this vine was 2.4-fold higher than the amount detected in vines infected with *D. seriata* 12 months post-inoculation. No target PMs were detected in the negative control vines. Targeted PMs were also not detected in lesion-free wood samples (AA’ and BB’, Figure 3), thus, no further analyses were performed for tissues collected further away from the necrotic wood (IP) including the leaves.

## 3. Discussion

To the best of our knowledge, this is the first study to investigate the production and translocation of PMs by Botryosphaeriaceae species in BD naturally infected and artificially inoculated vines using a multifaceted approach. However, multidisciplinary approaches for the detection of target PMs in plant tissues have been reported in other pathosystems [42,43,44,45].

This current study also represents the first study to use a combination of conventional plant pathology and molecular techniques to detect and quantify Botryosphaeriaceae DNA from artificially inoculated and naturally infected vines. The qPCR primers and probe used in this study [46] are not species-specific, therefore they cannot distinguish the Botryosphaeriaceae pathogen at the species level. However, the isolation of pathogens from naturally infected wood allowed the identification of Botryosphaeriaceae spp. Similarly, re-isolations of the pathogens from artificially inoculated vines resulted in the recovery of *D. seriata*, *S. viticola* and *Do. vidmadera* from previously inoculated vines. Lesion length by itself was insufficient to show differences in disease severity and pathogen virulence at the early stage of infection (6 months post-inoculation). Furthermore, negative control vines also exhibited necrosis near the inoculation point similar to those vines inoculated with pathogens. The qPCR analysis, on the other hand, was able to quantify the amount of pathogen DNA between treatments and assessment periods. Consequently, pathogen DNA was not detected in negative control vines. It is important to note that low levels of pathogen DNA were detected from lesion-free tissues adjacent to the lesions from a few inoculated vines at 6-month post-inoculation, particularly for *S. viticola*. This suggests that the pathogens were able to move endophytically beyond the lesions as latent pathogen, similar to the study reported by Billones-Baaijens et al. [47]. The absence of pathogen DNA in non-necrotic tissues adjacent to the lesions at 12 months post inoculation suggests that as the infection advanced, these pathogens shifted from being latent to necrotrophic.

The qPCR analysis of inoculated vines showed a significant difference in susceptibility between Chardonnay and Cabernet Sauvignon to different Botryosphaeriaceae species, with the former being more susceptible to *D. seriata*. The reduced susceptibility of Cabernet Sauvignon to some Botryosphaeriaceae species may be associated with the amount of stilbene polyphenols that are usually higher in red vine varieties. These compounds have fungistatic activity [48,49], that may assist the plant in limiting infection by BD pathogens. Nevertheless, stilbene polyphenols antimicrobial activity depends on the pathogen infecting the plant [48,49]. For instance, Cabernet Sauvignon is one of the most susceptible cultivars to Esca pathogens: *Phaeomoniella chlamydospora* and *Phaeoacremonium minimum* [10].

The qPCR analysis used in this study was also useful in quantifying pathogen DNA in symptomatic wood samples collected from naturally infected vines. Overall, these results confirm that molecular techniques could be applied to determine and quantify the pathogens in field material.

The LC-MS/MS analysis of the naturally infected and artificially inoculated vines showed that (*R*)-mellein can be detected in infected woody tissues with symptoms of BD. A previous study that investigated the production of Botryosphaeriaceae PMs in planta only used a limited number of naturally infected vines with both BD and Esca symptoms [34]. More recently (*R*)-mellein was also detected in grapevine tissues with symptoms of Esca and Grapevine leaf stripe disease (GLSD) [50]. However, none of the previous studies identified the pathogens or quantified the pathogen DNA in the analysed samples. Furthermore, the application of a multifaceted approach suggests a probable correlation between the amount of pathogen DNA in the wood and the area of the peak of the (*R*)-mellein in the chromatograms. This correlation was more evident in the Riverina vines, which resulted in more intense peaks corresponding to (*R*)-mellein and the highest amount of pathogen DNA in the wood. Moreover, the amount of (*R*)-mellein detected in the wood infected with *N. parvum*, an aggressive Botryosphaeriaceae pathogen [4], 6 months post-inoculation was 2.4-fold higher than the amount detected in vines infected with the lesser aggressive *D. seriata* at 12 months post-inoculation. These data further support those previously reported that the amount of (*R*)-mellein produced by *N. parvum* and *D. seriata* under in vitro conditions is proportional to the aggressiveness of the pathogens [51].

BD pathogens colonize the woody parts of the plants, and during colonization, the fungi can produce and release a series of PMs. The translocation hypothesis suggests that PMs produced by the pathogens can migrate far from the inoculum along the asymptomatic wood, reaching the leaves or the green shoots without being catabolized or without being entirely detoxified [26]. The main goal of our experiment was to verify the movement of targeted PMs along the trunk. For this purpose, symptomatic and adjacent lesion-free functional woody tissues (trunk and branches) were sampled. To ensure that the detection of the target PMs in asymptomatic wood was due to their migration, qPCR analysis was performed. Our results showed that (*R*)-mellein was detected only in the wood samples with necrotic lesions showing a high amount of pathogen DNA. The failure to detect (*R*)-mellein in all the lesion-free wood samples suggests that, at least under these experimental conditions, the translocation of (*R*)-mellein in its native form did not occur along the trunk to the foliage. It is possible that *D. seriata* produces (*R*)-mellein as part of its strategy to breakdown the host cells during infection. In a recent study using in vitro plantlets, Trotel-Aziz et al. showed that *(R)*-mellein strongly suppresses the expression of genes involved in plant defence and that *(R)*-mellein may be accumulated in planta in its native chemical form. The detection of this compound in symptomatic wood during our experiment supports this hypothesis [52].

However, the mere presence of PMs in the wood may not be enough to induce foliar symptoms in grapevines with BD. The development of symptoms may be more complicated than previously thought, and they may arise from interactions between biotic and abiotic stresses (water stress, drought, heat stress), which require more in-depth studies to understand. Fungi involved in GTDs can act as endophytes for several years before becoming pathogens, and many have hypothesised that the abiotic conditions, in particular thermal and water stress, can weaken the plant defence and therefore result in the development of GTD foliar symptom [9,10,22,53,54,55,56]. For instance, previous investigation, showed that GLSD leaf symptoms increased when the rainfall was abundant in June–July [54].

More comprehensive field studies on the influence of climatic conditions on foliar symptoms associated with GTD, including the role of PMs produced in planta, are fundamental for elucidating the relationship between fungal PMs and physiological changes in the vine which result in the expression of foliar symptoms.

Another result arising from this study was the failure to detect protocatechuic acid and spencertoxin in the LC-MS/MS analysis of vines inoculated with *D. vidmadera* and *S. viticola*, respectively. These two species were shown to produce these PMs in vitro [35,39]. The lack of detection of these PMs may be due to various reasons: (i) they may not be produced in planta; (ii) they could be detoxified by the plant; (iii) they may form toxin derivatives with other compounds; and/or (iv) they may be irreversibly bound to the wood contributing to lesion expression. The latter hypothesis was already suggested for other phytotoxins produced by the ED pathogen, *E. lata* [27,28,29]. Overall, the general conclusion may be that not all the secondary metabolites produced in vitro can be detected in planta since their fate mainly depends on the biological role played in the interaction of the pathogen with the host. This can be further validated by applying our approach to different experimental conditions. For instance, plant materials with reported BD foliar symptoms could be analyzed to further investigate the role of these PMs in foliar symptom development. Comparing data obtained from various vine-growing regions can be valuable to validate our results and help to explain the lack of BD foliar symptoms observed in Australian vineyards.

PMs are also known to interact with different cellular targets or can inhibit the activity of plant enzymes [57]. All these processes can result in the formation of toxin conjugates or derivatives that could explain the failure to detect target PMs in their native form. Metabolomics approaches can be applied to investigate the formation of PM derivatives and their role in symptoms development or plant-pathogen interaction. Indeed, untargeted, and targeted metabolomics have become fundamental tools in plant science and chemical ecology [58,59]. Recently, metabolic changes in grapevine wood infected with *N. parvum* have been reported [60]. This was accompanied by an accumulation of a number of unknown metabolites in the infected wood samples. Their structural elucidation will be crucial to better understand the response of the plant to GTDs and to also identify if any of these unknown metabolites could be potential derivatives of PMs.

## 4. Materials and Methods

### 4.1. Artificially Inoculated Vines

#### 4.1.1. Planting Materials

*Vitis vinifera* cvs. Chardonnay and Cabernet Sauvignon, two of the most commonly grown varieties in Australian vineyards [61] were selected for a glasshouse experiment. Twenty- eight dormant, apparently healthy cuttings for each variety were collected in a commercial vineyard in Hilltops, New South Wales (NSW), Australia in Winter (June 2017) and stored at 4 °C for 4 weeks until rooting. All cuttings were surface-sterilised with 0.5% sodium hypochlorite for 1 min, rinsed twice with tap water and rooted in plastic trays containing perlite. The trays were placed on heat mats at 30 °C for 4 weeks to facilitate rooting. The rootlings were planted in 10 L pots containing commercial garden mix (60% compost, 20% wash sand, 20% screen loam). All vines were maintained in a glasshouse (17–27 °C) and watered every 12 h for 5 min (8 L/h) with an automatic dripper system for 6–12 months until assessment.

#### 4.1.2. Fungal Isolates

The three Botryosphaeriaceae species: (a) *Diplodia seriata* H141a; (b) *Spencermartinsia viticola* DAR78870 and; (c) *Dothiorella vidmadera* DAR78993 from the National Wine and Grape Industry Centre (Charles Sturt University, Wagga Wagga, NSW, Australia) culture collection, which produced PMs in vitro and were previously characterised [33,34,37], were used for inoculating the glasshouse vines in Spring (November 2017). For inoculations, wounds were created in the middle internode of the trunk for each vine using a flame-sterilised 4 mm sterile cork borer. Mycelial plugs (4 mm) cut from the margins of 4- day-old cultures of the selected fungal species grown on potato dextrose agar supplemented with Chloramphenicol (100 mg/L) (PDA-C), were inserted into the wounds and sealed with Parafilm (Bemis, USA). Sterile non-colonised plugs of PDA-C were used as negative controls. The inoculated vines for each variety were arranged in a randomised complete block design (RCBD) at 7 replications per inoculum per variety combination.

#### 4.1.3. Sampling of Artificially Inoculated Vines

The trunks of randomly selected vines were cut at the base and shoots were trimmed off. The bark surrounding the inoculation point was removed and lesions were measured using a digital caliper (Workzone, Australia). Tissue samples from each vine were collected as shown in Figure 3. Three different samples were collected from each vine: (a) trunk sections with visible lesions including the inoculation point (IP); (b) 2 cm lesion-free trunk sections cut above and below the necrotic lesions (labelled as AA’); (c) 2 cm trunk sections above and below section A and A’ (labelled as BB’). The samples were surface-sterilised for 2 min in 70% ethanol and rinsed three times with sterile deionised water (SDW) before cutting longitudinally to obtain four quarters of each section. One quarter of the section was used for isolation of the pathogen, while the remaining sections were stored at −80 °C and used for DNA and toxin extractions.

#### 4.1.4. Fungal Isolation from Artificially Inoculated Vines

One quarter of surface-sterilised trunk sections collected from the inoculated vines (Figure 1) were placed onto PDA-C. Plates were incubated at 25 °C in the dark and observed for growth of Botryosphaeriaceae species for 4–7 days. The re-isolated pathogens were identified using morphological and molecular methods. Three vines per treatment were assessed at 6 months post inoculation in Autumn (May 2018). Four vines per treatment were further assessed at 12 months post inoculation which occurred in Spring (November 2018).

### 4.2. Fungal Isolation from Naturally Infected Vines

Wood samples (cordons and trunks) exhibiting BD cankers were collected from 20–24 years old grapevines from three wine regions in NSW, Australia in Winter (June–August 2017). The Tumbarumba region has a cool climate with an altitude of 700 m. The Hilltops region has a continental climate with an altitude of 450 m. The Riverina region has a semiarid climate with an altitude of 66–540 m [62]. Four to five wood pieces (10–20 cm) with necrotic lesions and non-necrotic tissues 20–30 cm away from the necrotic ones were cut from three different vines in Hilltops (cv. Shiraz), Riverina (cv. Shiraz) and Tumbarumba (cv. Chardonnay) and stored at 4 °C for 2 weeks until processed. Each sample were processed and analysed individually. For isolations, the bark was removed to expose the necrotic lesions and the wood was cut into ~1 cm sections (20–24 pcs) with each section containing necrotic and healthy wood. All sections were surface-sterilised following the methods described for the inoculated vines. Approximately 10–12 pieces of the surface-sterilised tissues were stored at −80 °C for DNA and toxin extractions. The remaining sections were plated onto PDA-C with four sections per plate for a total of three plates per vine. Plates were incubated at 25 °C for 4–7 days and observed for growth of Botryosphaeriaceae species. All Botryosphaeriaceae isolates were identified by morphological and molecular methods.

### 4.3. DNA Extractions from Fungal Mycelia

DNA samples were extracted from the mycelium for all the isolated fungi using PrepMan Ultra (Applied Biosystems, UK) and following the manufacturer’s instructions. Mycelium (~100 mg) was scraped from the edge of the colony for each isolate using a sterile pipette tip and transferred into a sterile 1.5 mL tube containing 100 μL of the PrepMan Ultra (Applied Biosystems, UK) preparation reagent. Mycelial suspensions were vortexed for 30 s and incubated at 95 °C in a heat block for 10 min. The tubes were centrifuged for 2 min at 3220× *g* and 50 μL of the supernatant was transferred to a new sterile 1.5 mL tube and stored at −20 °C until required for PCR.

### 4.4. DNA Extraction from Grapevine Wood

Wood samples stored at −80 °C were freeze-dried (Christ, John Morris Scientific, USA) for 24–36 h. Dried samples were homogenized at 20 Hz for 2 min 30 s using 10 mL grinding jars attached to a TissueLyser II (Qiagen, Hilden, Germany). The ground wood (100 mg) was transferred into a sterile 2 mL tube for DNA extraction while the remaining ground wood was stored in a separate tube for the extraction of toxins. DNA was extracted from wood samples using the methods described by Pouzoulet et al. [63] with some modifications. The CTAB (Cetyl Trimethyl Ammonium Bromide) extraction buffer was prepared according to Doyle and Doyle [64], and 1 ml was added to each tube containing 100 mg of ground wood and gently mixed by pipetting. The mixture was incubated at 65 °C for 1 h using a heat block. After incubation, 500 µL of chloroform/isoamyl alcohol (24:1, Sigma Aldrich, St. Louis, MI, USA) was added and the tube was inverted 10×, incubated on ice for 5 min and then centrifuged at 4 °C for 10 min at 2300× *g*. Approximately 420 µL of the lysate was pipetted into a QIAshredder spin column placed in a 2 mL collection tube from the Qiagen DNeasy Plant DNA extraction kit (Qiagen, Hilden, Germany) and centrifuged for 2 min at 20,000× *g*. The subsequent steps were performed using the buffers, materials and protocol from the DNeasy Plant DNA extraction kit. All DNA samples were eluted to a final volume of 100 µL using the Qiagen AE buffer. All DNA samples were quantified using a Quantus™ Fluorometer (Promega, Madison, WI, USA) prior to qPCR.

DNA was further extracted from the necrotic tissues collected from potted vines (cv. Chardonnay) inoculated six months prior with *N. parvum* DAR78998 following the methods described above. This isolate was found to be highly virulent in a separate experiment [18] and is known to produce (*R*)-mellein in vitro [35]. The DNA extracted from these vines was included in the qPCR analysis and toxin analyses.

### 4.5. Identification of Isolated Botryosphaeriaceae by PCR

All Botryosphaeriaceae recovered from the artificially inoculated and naturally infected vines were identified using PCR and DNA sequencing. To amplify the internal transcribed spacer (ITS) region of the ribosomal DNA of the pathogens, PCR was performed using universal primers ITS1 and ITS4 [65]. Each 25 µL PCR reaction contained 1x PCR buffer (Bioline, Memphis, TN, USA), 0.4 µL for each of primer, 1.25 U of My TaqRed DNA polymerase (Bioline, Memphis, USA) and approximately 1–5 ng of DNA template. PCRs were performed using a thermal cycler (C100 Thermal cycler, Biorad Laboratories, Pty, Ltd., Hercules, CA, USA) under the following conditions: initial denaturation at 95 °C for 5 min, 35 cycles of 30 s at 94 °C, 45 s at 55 °C, and 90 s at 72 °C, with a final extension of 5 min at 72 °C. Following amplification, the PCR products were visualized by gel electrophoresis. PCR products were purified with FavorPrep Gel/PCR purification kit (Favorgen Biotech Corp, Taiwan) and sequenced at the Australian Genome Research Facility (AGRF; Sydney, NSW, Australia). All DNA sequences and chromatographs were analyzed using the DNAMAN 5.2 (Lynnon Biosoft©, San Ramon, CA, USA) and Chromas Lite 2.1© (Technelysium PTY Ltd, Brisbane, Australia) software. All trimmed DNA sequences were analysed using the Basic Local Alignment Search Tool (BLAST) in GenBank (https://www.ncbi.nlm.nih.gov/genbank/ (accessed on 18 April 2021)).

### 4.6. Quantification of Botryosphaeriaceae spp. from Wood Samples by qPCR

The qPCR assay using Botryosphaeriaceae multi-species primers and hydrolysis probe developed by Billones-Baaijens et al. [46] were used to detect and quantify Botryosphaeriaceae spp. from artificially inoculated and naturally infected vines. All qPCR assays were performed with the RotorGene 6000 system (Corbett Life Science, Qiagen, Hilden, Germany) using Botryosphaeriaceae multi-species primers Bot-BtF1 (5’-GTATGGCAATCTTCTGAACG-3’) and Bot-BtR1 (5’-CAGTTGTTACCGGCRCCRGA-3’), and a hydrolysis probe, Taq-Bot probe 5’-/56-FAM/TCGAGCCCG/ZEN/GCACSATGGAT/3IBkFQ/-3’) [41]. For each assay, three controls were included: (1) non-template control (H_2_O); (2) standard (500 pg) Bot-Btub gBlock [41]; (3) DNA from vine inoculated with *N. parvum* DAR78998 at four technical replicates each.

For the artificially inoculated vines, all necrotic tissue samples collected from the IP were first analysed followed by the tissues collected above and below the necrotic lesions (AA’). For AA’ tissues which tested positive to Botryosphaeriaceae DNA, their subsequent BB samples were further analysed for a total of 32 and 36 samples for 6- and 12- months PI, respectively. For naturally infected vines, wood samples containing necrotic tissues were first analysed by qPCR. When the necrotic tissue sample was positive to qPCR, the healthy wood samples away from the necrotic tissue were further analysed for a total of 18 samples overall.

To determine the amount of pathogen DNA that was amplified by each qPCR assay, previously developed standard curves [46] were imported in the Rotor-Gene Q software (Version 2.3.1). The standard (Bot-Btub gBlock, 500 pg) that was included in each qPCR assay was used to calibrate the imported standard curve and the resulting regression equations were used to quantify the number of Botryosphaeriaceae β-tubulin gene copies in each reaction as previously described by Billones-Baaijens et al. [46] following the MIQE guidelines [66]. To calculate the number of copies of the Botryosphaeriaceae β-tubulin gene in each wood sample, the following formula was used:N = g (d × c)/t × c,(1)
where N: calculated number of β-tubulin gene copies in one wood sample, g: the mean number of gene copies detected by qPCR; d: total gDNA extracted from 100 mg of wood (100 µL), c: DNA concentration (µL); T: the amount of DNA template (5 µL) in one reaction.

### 4.7. Statistical Analysis

Data arising from the glasshouse experiment were analysed using IBM SPSS 24 software. All data were tested for homogeneity using Levene’s test at *p* ≤ 0.05. For inoculated vines, univariate analysis of variance (ANOVA) was used to assess differences in lesion lengths and pathogen copies between varieties, inoculated pathogens and their interactions (*p* ≤ 0.05). All means were separated by pairwise comparison using Fischer’s least significant differences (LSD) test at 5% significance level. Statistical analysis was not applied to the naturally infected vines because of differences in the storage and sampling of the collected wood.

### 4.8. Chemicals and Standards for LC-MS/MS

CH_3_Cl, MeOH and *n*-hexane were analytical grade (Sigma-Aldrich, St. Louis, USA). H_2_O, 0.1% HCOOH and MeCN were LC-MS grade (Sigma-Aldrich, St. Louis, USA). The (*R*)-mellein [35], protocatechuic alcohol [36] and spencertoxin [39] that were used as standards were isolated from in vitro cultures of *D. seriata*, *Do. vidmadera* and *S. viticola* as previously reported.

### 4.9. Testing of Protocols for Extraction of PMs from Wood

Two published extraction protocols were compared for their suitability to extract toxins from wood samples [34,41]. Freeze-dried wood samples collected from the naturally infected vines (36 in total) were extracted using both protocols and were analysed by LC-MS/MS. Freeze-dried wood samples (100 mg) were used for each extraction. The protocol by Saviano et al. [41] was subsequently used to extract PMs from the wood samples (100 mg) collected from inoculated vines (100 in total).

### 4.10. LC-MS/MS Analysis of Targeted PMs from Wood

Analyses were carried out using a 1290 Infinity II LC system (Agilent) hyphenated to an Agilent 6470 triple quadrupole (QqQ). The UPLC system included a binary pump and a cooled autosampler maintained at 15 °C. Mass Hunter software was used to control the instruments and to acquire the data which were then processed for analysis. The chromatographic separation was performed using a reverse phase column Phenomenex Luna 5.4 µm 250 × 4 mm i.d., protected by a security guard column Phenomenex maintained at 30 °C. The mobile phase consisted of H_2_O 1% (*v*/*v*) HCOOH (Phase A) and MeCN (Phase B). The flow rate was 0.7 mL/min, the gradient system was initiated with 10% of Phase B for 2 min and reached 30% at 15 min, 80% at 25 min, isocratic until 27 min and 95% at 40 min. Samples were injected into the column with an injection volume of 20 µL.

The Agilent 6470 triple quadrupole (QqQ) was used as the detector in MRM mode with electrospray ionization (ESI) in positive ionization mode. The source and desolvation temperatures were respectively set at 350 °C, Nebulizer, 40 psi; N_2_ flow, 12 L min^−1^. Capillary voltage was set at 3.5 kV in positive mode. The MRM transitions (precursor ion → daughter ions), fragmentor energy and collision energy for *(R)*-mellein, spencertoxin and protocatechuic acid were optimized using the Agilent Optimizer Software, and the optimized parameters were shown in Table 2. For the parameter optimization, standards of (*R*)-mellein, spencertoxin and protocatechuic acid were used.

## 5. Conclusions

Detection of PMs in infected plant tissue may provide insights into the involvement of PMs in the pathogenicity of fungal isolates and symptom development of GTDs. This multidisciplinary approach was appropriate for investigating the production and translocation of (*R*)-mellein in vines showing BD symptoms. Our results highlighted a possible correlation between the amount of (*R*)-mellein and the amount of pathogen DNA copies in the wood samples. The development of a robust quantitative LC-MS/MS method for the detection of (*R*)-mellein can assist further investigations regarding this correlation. These data may provide valuable information on the infection strategies of the pathogens and may clarify the role played by (*R*)-mellein in the development of the disease.

Our study did not find any evidence that PMs are translocated into healthy tissues of the vine. Thus, the targeted PMs we investigated in this study were most likely produced by BD pathogens for penetrating and invading the hosts. The migration of PMs in their native form into the woody tissues of the plant should not be considered a generalised process for all pathosystems.

## Figures and Tables

**Figure 1 plants-10-00802-f001:**
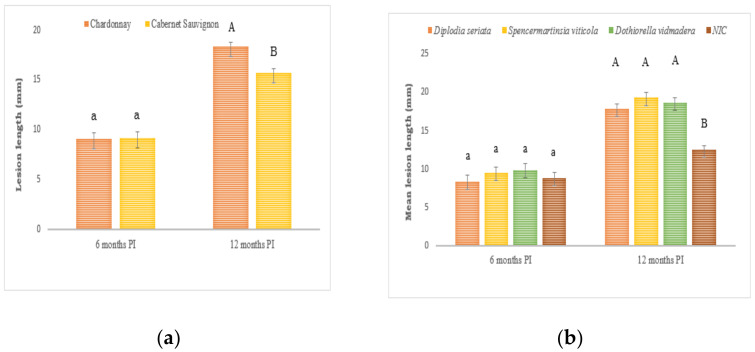
(**a**) Overall mean lesion lengths on Chardonnay and Cabernet Sauvignon at 6- and 12-months (PI). (**b**) Mean lesion caused by *Diplodia seriata* H141a, *Dothiorella vidmadera* DAR78993 and *Spencermartinsia viticola* DAR78870 at 6- and 12-months PI = post inoculation. NIC = non-inoculated control. Lowercase letters refers to 6 months PI vines, uppercase letters refer to 12 months PI vines. Bars with different letters for each inoculation period are significantly different at *p* ≤ 0.05 least significant difference (LSD). Error bars are standard error of the means.

**Figure 2 plants-10-00802-f002:**
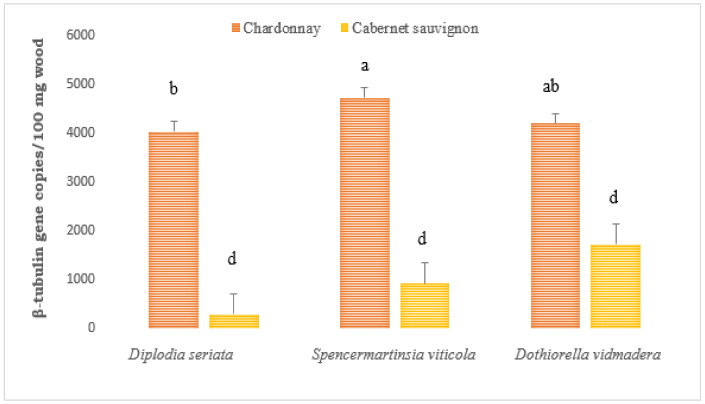
Overall mean of Botryosphaeriaceae β-tubulin gene copies detected from *Vitis vinifera* (cvs. Chardonnay and Cabernet Sauvignon) inoculated with *Diplodia seriata* H141a, *Spencermartinsia viticola* DAR78870 and *Dothiorella vidmadera* DAR78993 at 6 months post inoculation using quantitative PCR. Bars with different letters for each inoculation period are significantly different at *p* ≤ 0.05 LSD. Error bars are standard error of the means. All non-inoculated control vines tested negative to Botryosphaeriaceae DNA and were excluded in the graph.

**Figure 3 plants-10-00802-f003:**
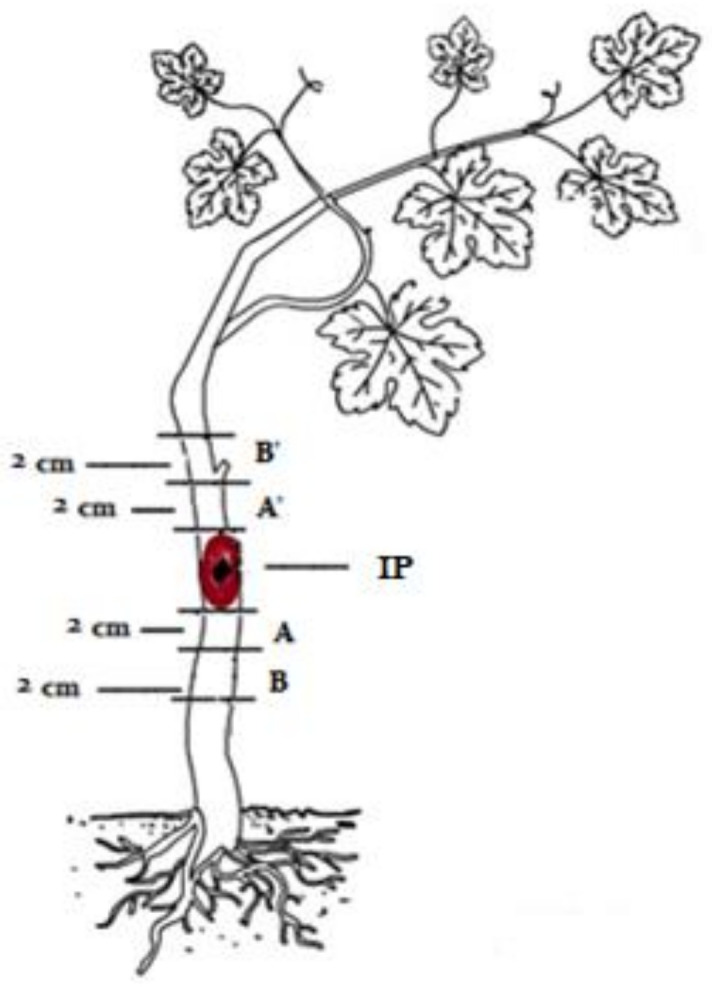
A diagram of an inoculated vine showing the positions and sizes of tissue samples collected and used for analysis. IP: inoculation point. AA’: lesion-free trunk sections subsequent to necrotic lesions; BB’: lesion-free trunk sections subsequent to AA’.

**Figure 4 plants-10-00802-f004:**
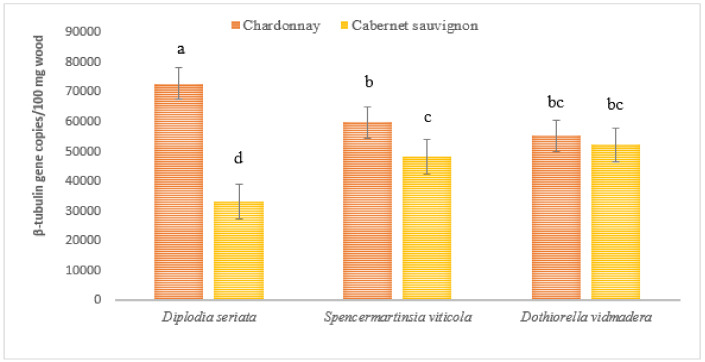
Overall mean of Botryosphaeriaceae β-tubulin gene copies detected from necrotic tissues at the inoculation point of *Vitis vinifera* (cvs. Chardonnay and Cabernet Sauvignon) vines inoculated with *Diplodia seriata* H141a, *Spencermartinsia viticola* DAR78870 and *Dothiorella vidmadera* DAR78993 at 12 months post inoculation using quantitative PCR. Bars with different letters for each inoculation period are significantly different at *p* ≤ 0.05 LSD. Error bars are standard error of the means. All non-inoculated vines tested negative to Botryosphaeriaceae DNA and were excluded in the graph.

**Figure 5 plants-10-00802-f005:**
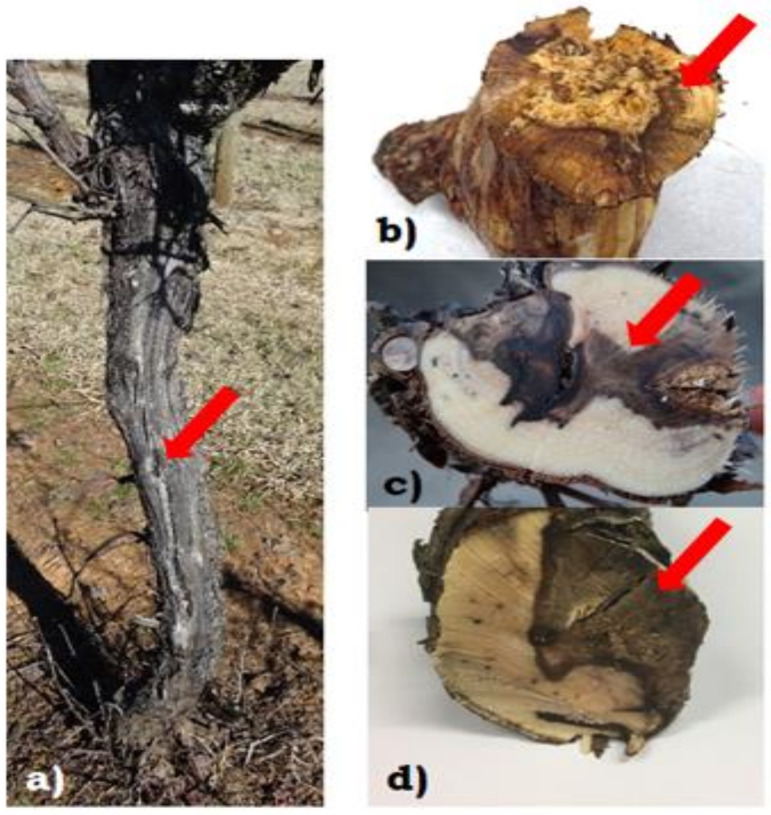
(**a**) Vine with trunk canker (arrow); (**b**) cross-section of the trunk with central necrosis; (**c**) cross-section of the cordon with wedge-shape necrosis; (**d**) cross-section of the trunk with wedge-shape necrosis.

**Figure 6 plants-10-00802-f006:**
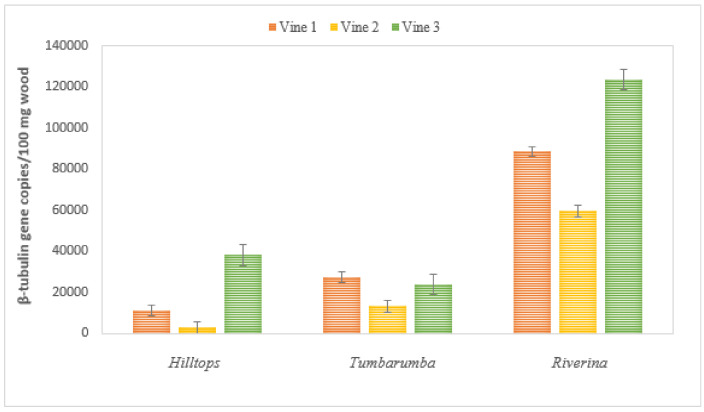
Number of copies of pathogen DNA quantified by qPCR in naturally infected vines from three vineyards in New South Wales, Australia. Error bars are standard deviation of the means.

**Table 1 plants-10-00802-t001:** Botryosphaeriaceae species isolated from naturally infected vines from three vineyards in New South Wales, Australia.

Location	Variety	Vine Sample	Botryosphaeriaceae Species
Hilltops	Chardonnay	1	*Neofusicoccum parvum*
		2	*Diplodia seriata*
		3	*Diplodia mutila*
Tumbarumba	Chardonnay	1	*D.seriata*
		2	*D. seriata,* *N. parvum*
		3	*D. seriata*
Riverina	Shiraz	1	*D. seriata*
		2	*Botryosphaeria dothidea*
		3	*D. seriata*

**Table 2 plants-10-00802-t002:** Toxins used as standards for the LC-MS/MS and their corresponding optimized parameters.

Toxin	Precursor Ion m/z	Fragment Ion m/z	Fragmentor Voltage *	CV **	Retention Time (Min)
*(R)*-mellein 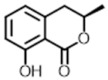	179.1 [M + H]^+^	161.0 133.0 105.0	90	12 16 24	27.44
Protocatechuic alcohol 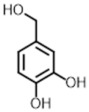	123.1 [M-H2O + H]^+^	67.1 55.1 51.1	90	16 24 40	6.94
Spencertoxin 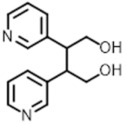	283.1 [M + K]^+^	177.8 118	90	44 50	18.11

* Fragmentor voltage: controls the speed at which the ions pass through a medium pressure capillary between the electrospray chamber and the mass spectrometer. ** CV: collision energy voltage.

## Data Availability

The data presented in this study are contained within the article and Appendix A.

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
