# Peer review of "Production of Phytotoxic Metabolites by Botryosphaeriaceae in Naturally Infected and Artificially Inoculated Grapevines"

_plants, 2021, doi:10.3390/plants10040802_

Round 1
Reviewer 1 Report
The publication describes a study on the role of phytotoxic metabolites in the expression of Botryosphaeria dieback symptoms in naturally-infected and artificially-inoculated wood using molecular and analytical chemistry techniques.
Detailed comments:
Abstract
Lines 19-20: “Foliar symptoms in vines affected with Botryosphaeria dieback (BD) have not been reported in Australian vineyards to date”. It is not relevant information for the abstract.
Introduction
Line 38: Include a more current bibliographic citation. For example: Sancho-Galán, P.; Amores-Arrocha, A.; Jiménez-Cantizano, A.; Palacios, V. Influence of the Presence of Grape Skins during WhiteWine Alcoholic Fermentation. Agronomy 2021, 11, 452. https://doi.org/10.3390/ agronomy11030452.
The objectives of the manuscript are not clear in the introduction. I suggest that you review the text between lines 71 and 85.
Results
Line 92. Include pictures of the cross sections of the trunks as a supplementary figure.
Line 110-114: The authors affirmed that: “There were no significant interactions between varieties and inoculation treatments based on lesion lengths”. For it, they should be shown the mean lesion caused by Diplodia seriata H141a, Dothiorella vidmadera DAR78993 and Spencermartinsia viticola 114 DAR78870 at 6 and 12 months PI per cultivar. Figure 2 does not indicate to which cultivar the data correspond.
Figures 1, 7-9, could be moved to supplementary figures.
Material and methods
For future experiments, you should be considered to disinfect the plant material to ensure the reliability of the results. Some studies have proposed immersing the plant material in hot water as an effective technique to prevent the propagation of fungi.
Line 409: The authors should include the coordinates of the vineyards locations where the samples were collected.
Line 508: There is writing mistake.
Reviewer 2 Report
I found the manuscript clear in its intentions and well prepared; the topic is worthy of dissertation, as well as the amount of results and their discussion leave the reader interested and satisfied.
I have only a few minor issues to report:
Line 119: "point (IP; Figure 1) from vines inoculated with all three Botryosphaeriaceae species." change with "point (IP) from vines inoculated with all three Botryosphaeriaceae species (Figure 1)"
Line 121: "with significantly higher (P<0.05) amounts of Botryosphaeriaceae DNA than Chardonnay inoculated with D. seriata." change with "with significantly higher amounts of Botryosphaeriaceae DNA than Chardonnay inoculated with D. seriata (P<0.05) "
Line 196 and 200: "organic extracts" change with "organic compounds"
I found Figures 7, 8 and 9 not necessary in the main text; Authors are invited in moving them to supplementary materials
4.2: please provide the Collection of provenience of fungal strains
4.5: "Mycelia" change with "Mycelium"
Line 420: delete "Crude"
Reviewer 3 Report
Letter for Authors – “Production of phytotoxic metabolites by Botryosphaeriaceae in naturally-infected and artificially-inoculated grapevines” – plants-1171663 - Reviewer 33
Dear Authors,
I found your manuscript interesting for the possibilities to extend the knowledge about GTD reporting evidence on a well not known aspect as the role of Botryosphaeriaceae phytotoxic metabolites in grapevine.
However the manuscript needs some corrections, suggested below.
Introduction
Line 37: replace “Grapevines are one….crops” with “Grapevine is one….crop”;
Line 45: [6,7] references. These are two reviews that report the same topics. My suggestion is to replace one of the two with a research article reporting data on GTD incidence and related pathogens, focused in particularly on Botryosphaeriaceae:
Baránek, M.; Armengol, J.; Holleinová, V.; Pečenka, J.; Calzarano, F.; Peňázová, E.; Vachůn, M.; Eichmeier, A. Incidence of symptoms and fungal pathogens associated with grapevine trunk diseases in Czech vineyards: First example from a north-eastern European grape-growing region. Phytopathol. Mediterr. 2018, 57, 449–458, ISSN: 0031-9465, doi:10.14601/Phytopathol_Mediterr-22460.
Lines 45-47: this sentence is rather reductive, because it does not take into account the relevant findings recently obtained on GLSD, the most common disease of Esca complex, regarding symptom expression mechanisms and management, and phytotoxic metabolites. Therefore, I suggest to add this sentence, and related suggested references, before the sentence at line 45: “Recently, significant findings have been obtained on Grapevine Leaf Stripe Disease, a widely spread wood disease of Esca complex, regarding phytotoxic metabolites, symptom expression and their management [8,9].”
- Calzarano, F.; D’Agostino, V.; Pepe, A.; Osti, F.; Della Pelle, F.; De Rosso, M.; Flamini, R.; Di Marco, S. Patterns of phytoalexins in the grapevine leaf stripe disease (esca complex)/grapevine pathosystem. Phytopathol. Mediterr. 2016, 55, 410–426. ISSN: 0031-9465, DOI: 10.14601/Phytopathol_Mediterr-18681
- Calzarano, F.; Di Marco, S. Further evidence that calcium, magnesium and seaweed mixtures reduce grapevine leaf stripe symptoms and increase grape yield. Phytopathol. Mediterr. 2018, 57, 459-471, ISSN: 0031-9465, DOI: 10.14601/Phytopathol_Mediterr-23636
Line 47: [6,7,9,10]: delete one of the two references of the reviews.
Lines 48-49: Please, add “…diseases” after “Esca complex….”.
Materials and Methods
Line 362: “4.1. Artificially Inoculated Vines” is the subsection title. The subsequent paragraphs are subsubsections. You could title the subsubsection from line 363 to line 372: “4.1.1. Experimental vines”, or you can choose an appropriate title. Therefore, the following subsubsections will be: “4.1.2. Fungal Isolates Used for Inoculation”, and “4.1.3. Fungal Isolation from Inoculated Vines”.
Line 364: “[57]” wrong reference?; “(Wine Australia 2018)” delete
Line 382-384: unclear; You have produced 15 vines per cultivar. Wouldn’t you need 21 vines per cultivar to inoculate the three fungi (7 replicates)?
Line 396, 414, 432: maybe you could move this information in the related subsections, DNA and toxin extraction.
Line 428: ..both from naturally-infected and artificially inoculated vines? Specify.
Line 510: This is an unusual statement. Apparently, it would mean that you can’t compare these data. Can you explain why you think they are valid?
Line 520: “…wood samples collected from the naturally-infected vines”. Also from artificially inoculated vines?
Line 525: “LC-MS/MS Analysis”, maybe you can add “…of PMs from wood”.
Results
Also in this section I suggest changes of section and subsection titles, as described below:
“2.1. Artificially-Inoculated Vines”
“2.1.1. Wood Symptoms”
“2.1.2. Botryosphaeriaceae DNA in wood tissues of artificially inoculated vines” instead of “qPCR Analysis of Symptomatic and Asymptomatic Wood Tissues from Inoculated Vines”
“2.3. Naturally-Infected Vines”, ok
“2.4. Botryosphaeriaceae DNA in wood tissues of naturally-Infected vines” instead of “qPCR Analysis of Naturally-Infected Wood Samples”
“2.5. Selection of Protocol for Extraction of PMs from Wood”
“2.6. PMs in wood tissues of naturally-Infected vines” instead of “LC-MS/MS Analysis of Naturally-Infected Wood Samples”
“2.7. PMs in wood tissues of artificially inoculated vines” instead of “LC-MS/MS Analysis of Inoculated Wood Samples”.
Line 127-128: it seems to me a repetition of what was said in lines 125-126. Please, verify, also at lines 155-159.
Line 140: from AA’ samples?
Line 171: is Table 2?
Line 181: Table 2: Please, do not use abbreviations.
Line 189-190: look at the above for line 510
Discussion
Line 275-277: critical, because Cabernet Sauvignon is one of the most susceptible cultivars to esca pathogens. It is also true that BD and esca are different diseases, but maybe you could point out this difference.
Line 327: You must consider that, in the article cited as reference 49, GLSD leaf symptoms increased when the June-July rainfall were abundant. Therefore, consider this evidence in what you write.
Line 350: delete
References
Line 692: Reference 49 shows incorrect the name of one of the Authors and the year of publication: the name of the Author is “Osti F.” and not “Fabio O.”, and the year of publication is “2018” and not “2019”; see below the correct reference:
Calzarano, F.; Osti, F.; Baránek, M.; Di Marco, S. Rainfall and temperature influence expression of foliar symptoms of grapevine leaf stripe disease (esca complex) in vineyards. Phytopathol. Mediterr. 2018, 57, 488-505, ISSN: 0031-9465, DOI: 10.14601/Phytopathol_Mediterr-23787
Finally, after considering my suggestions the manuscript can be published on Plants.
